

**A large role of missing volatile organic compounds reactivity**
**from anthropogenic emissions in ozone pollution regulation**
Wenjie Wang[1,2]*, Bin Yuan[1]*, Hang Su[2], Yafang Cheng[2], Jipeng Qi[1], Sihang Wang[1],
Wei Song[3], Xinming Wang[3], Chaoyang Xue[2], Chaoqun Ma[2], Fengxia Bao[2], Hongli
Wang[4], Shengrong Lou[4], Min Shao[1]
[1] Institute for Environmental and Climate Research, Jinan University,
Guangzhou 511443, China
[2] Multiphase Chemistry Department, Max Planck Institute for Chemistry, Mainz
55128, Germany
[3] State Key Laboratory of Organic Geochemistry, Guangzhou Institute of
Geochemistry, Chinese Academy of Sciences, Guangzhou 510640, China
[4] State Environmental Protection Key Laboratory of Formation and Prevention of
Urban Air Pollution Complex, Shanghai Academy of Environmental Sciences,
Shanghai 200233, China
*Correspondence to:* Bin Yuan (byuan@jnu.edu.cn);

18                      Wenjie Wang (Wenjie.Wang@mpic.de)



**Abstract:** There are thousands of VOC species in ambient air, while existing techniques
can only detect a small part of them (~ several hundred). The large number of
unmeasured VOCs prevents us from understanding the photochemistry of ozone and
aerosols in the atmosphere. The major sources and photochemical effects of these
unmeasured VOCs in urban areas remain unclear. The missing VOC reactivity, which
is defined as the total OH reactivity of the unmeasured VOCs, is a good indicator to
constrain the photochemical effect of unmeasured VOCs. Here, we identified the
dominant role of anthropogenic emission sources in the missing VOC reactivity
(accounting for up to 70%) by measuring missing VOC reactivity and tracer-based
source analysis in a typical megacity in China. Omitting the missing VOC reactivity
from anthropogenic emissions in model simulations will remarkably affect the
diagnosis of sensitivity regimes for ozone formation, overestimating the degree of
VOC-limited regime by up to 46%. Therefore, a thorough quantification of missing
VOC reactivity from various anthropogenic emission sources is urgently needed for
constraints of air quality models and the development of effective ozone control
strategies.




## 1 Introduction

Volatile organic compounds (VOCs) are key precursors of major photochemical pollutants, including ozone ($O_3$) and secondary organic aerosols(Atkinson, 2000;Atkinson and Arey, 2003). Severe $O_3$ and particle pollution are frequently related to high emissions of VOCs (Atkinson and Arey, 2003;Monks et al., 2015). There exist thousands of VOC species in ambient air that are emitted from either natural processes or anthropogenic activities (Goldstein and Galbally, 2007). No one instrument can capture all VOCs out there and even when they can be measured there is information missing on identification and properties (Yuan et al., 2017;Wang et al., 2014). As a result, the total amount of VOCs in ambient air has generally been underestimated. By now, emission inventories of VOCs used in air quality models only include the VOC species that can be measured, which will lead to an underestimation of the photochemical effect of total VOCs and thus causes uncertainties in predicting secondary pollution. The quantification of the unmeasured VOCs is crucial to assess secondary pollution precisely.

The measurement of total OH reactivity ($R_{OH}$) provides an effective approach to quantify the total amount of reactive gases in terms of reacting with OH radicals. The total OH reactivity is defined as:

$$R_{OH} = \sum_i k_{OH+Xi} [X_i], \tag{1}$$

where $X$ represents a reactive species including carbon monoxide (CO), nitrogen oxides ($NO_X$) and VOCs etc., and $k_{OH+Xi}$ is the reaction rate constant for the oxidation of species $X$ by OH. The measured $R_{OH}$ is higher than that calculated based solely on the measured reactive species, and the difference between them is mostly from unmeasured VOCs (Yang et al., 2017). Missing VOC reactivity (missing $VOC_R$), defined as VOC reactivity ($VOC_R$) of all unmeasured VOCs, can be obtained by subtracting the calculated $R_{OH}$ from the measured $R_{OH}$.

$$missing\ VOC_R = measured\ R_{OH} - calculated\ R_{OH} \tag{2}$$

$$calculated\ R_{OH} = \sum_i k_{OH+reactive\ species_i} [reactive\ species_i] \tag{3}$$



where reactive species represents measured VOCs and reactive inorganic species
including carbon monoxide (CO), nitric oxide (NO), nitrogen dioxide ($NO_2$), $O_3$, sulfur
dioxide ($SO_2$) and so on. The missing $VOC_R$ provides a constraint for evaluating the
photochemical roles of unmeasured VOCs in the atmosphere (Sadanaga et al.,
2005;Yang et al., 2016b). The inclusion of the missing $VOC_R$ can help to improve the
model performance in simulating photochemistry processes. Relatively high missing
$VOC_R$ have been found in forests (Di Carlo et al., 2004;Hansen et al., 2014;Nakashima
et al., 2014;Nölscher et al., 2016;Praplan et al., 2019), urban areas (Shirley et al.,
2006;Yoshino et al., 2006;Dolgorouky et al., 2012;Yang et al., 2017) and suburban areas
(Kovacs et al., 2003;Yang et al., 2017;Fuchs et al., 2017;Lou et al., 2010), accounting
for 10-75% of total $R_{OH}$.

The potential sources of missing $VOC_R$ include anthropogenic emissions, biogenic

emissions, soil emissions, and photochemical production, etc. (Yang et al., 2016b).
Previous studies have reported that the missing $VOC_R$ in forest areas was mainly from
either direct emissions or photochemical oxidation of biogenic VOCs (Di Carlo et al.,
2004;Hansen et al., 2014;Nakashima et al., 2014;Nölscher et al., 2016;Praplan et al.,
2019). Nevertheless, the dominant source of the missing $VOC_R$ in urban and suburban
areas remains unclear or under debate.

Surface $O_3$ pollution has become a major public health concern in cities worldwide

(Paoletti et al., 2014;Lefohn et al., 2018). A critical issue in determining an emission
control strategy for ozone pollution is to understand the relative benefits of NOx and
VOC emission controls. This is generally framed in terms of ozone precursor sensitivity,
i.e., whether ozone production is $NO_X$-limited or VOC-limited (Kleinman,
1994;Sillman et al., 1990). Nevertheless, the effect of missing VOCs on ozone
precursor sensitivity has not been well understood yet. Given that the missing $VOC_R$
accounts for a large part of total $VOC_R$, clearly clarifying the role of missing $VOC_R$ in
determining ozone precursor sensitivity is an urgent need for the diagnosis of ozone
sensitivity regimes and formulation of an effective emission reduction roadmap.

China has become a global hot spot of ground-level ozone pollution in recent years



(Lu et al., 2018;Wang et al., 2022). Pearl River Delta (PRD) remains one of the most
$O_3$-polluted regions in China (Li et al., 2022), although many control measures have
been attempted. Here, we measured $R_{OH}$ in Guangzhou, a megacity in PRD and
quantified the missing $VOC_R$. The dominant source of the missing $VOC_R$ and its impact
on ozone precursor sensitivity were comprehensively investigated.
**2 Method**
**2.1 $R_{OH}$ measurement**
The field campaign was conducted from 25 September to 30 October 2018 at an
urban site in downtown Guangzhou (113.2ºE, 23ºN). This site is primarily influenced
by industrial and vehicular emissions.
Total $R_{OH}$ was measured by the comparative reactivity method (CRM) (Sinha et
al., 2008). The CRM system consists of three major components, namely an inlet and
calibration system, a reactor, and a measuring system. Here, pyrrole ($C_4H_5N$) was used
as the reference substance in CRM and its concentration was quantified by a quadrupole
proton-transfer-reaction mass spectrometer (PTR-MS) (Ionicon Analytik GmbH,
Innsbruck, Austria). The CRM system was calibrated by propane, propene, toluene
standards and 16 VOC mixed standard (acetaldehyde, methanol, ethanol, isoprene,
acetone, acetonitrile, methyl vinyl ketone, methyl ethyl ketone, benzene, toluene, o-
xylene, α-pinene, 1,2,4-trimethylbenzene, phenol, m-cresol and naphthalene).
Measured and calculated $R_{OH}$ agreed well within 15% for all calibrations. The $R_{OH}$
measurement by the CRM method is interfered from ambient nitric oxide (NO), which
produces additional OH radicals via the reaction of $HO_2$ radicals with NO(Sinha et al.,
2008). To correct this interference, a series of experiments were conducted by
introducing different levels of NO (0–100 ppb) and given amounts of VOC into the
CRM reactor. A correction curve was acquired from these NO interference experiments,
which can be used to correct the $R_{OH}$ thank to the simultaneous measurement of ambient
NO concentrations. The detection limits of the CRM method were around 2.5 s$^{-1}$, and



the total uncertainty was estimated to be about 15%. The CRM method has been

successfully applied to measure OH reactivity in urban areas with high NO$_X$ levels in

previous studies (Dolgorouky et al., 2012;Yang et al., 2017;Hansen et al., 2015). The

intercomparison between the CRM method and pump–probe technique indicates that

the CRM method can be used under high-NO$_X$ conditions (NO$_X$>10 ppb) if a NO$_X$ -

dependent correction is carefully applied (Hansen et al., 2015).

**2.2 VOCs measurements**

Nonmethane hydrocarbons (NMHCs) were measured using a gas chromatograph–

mass spectrometer/flame ionization detector (GC–MS/FID) system coupled with a

cryogen-free preconcentration device(Wang et al., 2014). The system contains two-

channel sampling and GC column separation, which is able to measure C2–C5

hydrocarbons with the FID in one channel and measure C5–C12 hydrocarbons using

MS detector in the other channel. A total of 56 NMHCs species were measured. The

time resolution of the measurement was 1 h. The uncertainties of VOC measurements

by GC–MS/FID are in the range of 15 %–20 % (Wang et al., 2014;Yuan et al., 2012).

An online proton-transfer-reaction time-of-flight mass spectrometer (PTR-ToF-

MS) (Ionicon Analytic GmbH, Innsbruck, Austria) with H$_3$O$^+$ and NO$^+$ ion sources was

also used to measure VOCs. PTR-ToF-MS technique is capable of measuring

oxygenated VOCs (OVOCs) and higher alkanes that GC–MS/FID can not measure

(Wang et al., 2020a;Wu et al., 2020). The time resolution of PTR-ToF-MS

measurements was 10 s. A total of 31 VOCs were calibrated using either gas or liquid

standards. For other measured VOCs, we used the method proposed by Sekimoto et al.

(2017) to determine the relationship between VOC sensitivity and kinetic rate constants

for proton transfer reactions of H$_3$O$^+$ with VOCs. The fitted line was used to determine

the concentrations of those uncalibrated species. The uncertainties of the concentrations

for uncalibrated species were about 50 % (Sekimoto et al., 2017). The PTR-ToF-MS is

capable of measuring additional VOC species that GC–MS/FID cannot measure

including NMHCs with more carbons (C12–C20) and OVOCs including aldehydes,





ketones, carboxylic acids, alcohols and nitrophenols. Formaldehyde (HCHO) was
measured by a custom-built instrument based on the Hantzsch reaction and absorption
photometry (Xu et al., 2022).

**2.3 Other measurements**

Nitrous acid (HONO) was measured by a custom-built LOPAP (Long Path
Absorption Photometer) based on wet chemical sampling and photometric detection
(Yu et al., 2022). The uncertainty of the measurement was 8 %. $NO_X$, $O_3$, sulfur dioxide
($SO_2$) and CO were measured by $NO_X$ analyzer (Thermo Scientific, Model 42i), $O_3$
analyzer (Thermo Scientific, Model 49i), $SO_2$ analyzer (Thermo Scientific, Model 43i)
and CO analyzer (Thermo Scientific, Model 48i), respectively. The meteorological data,
including temperature (T), relative humidity (RH) and wind speed and direction (WS,
WD) were recorded by Vantage Pro2 Weather Station (Davis Instruments Inc., Vantage
Pro2) with a time resolution of 1 min. Photolysis frequencies of $O_3$, $NO_2$, HONO, $H_2O_2$,
HCHO and $NO_3$ were measured by a spectrometer (Focused Photonics Inc., PFS-100)
(Shetter and Müller, 1999;Wang et al., 2019).

**2.4 Multiple linear regression**

The multiple linear regression (MLR) have been successfully applied to quantify
the sources of air pollutants (Li et al., 2019;Yang et al., 2016a). In this study, a tracer-
based MLR analysis was used to decouple the individual contributions of
anthropogenic emissions, secondary production, biogenic emissions and background
level to missing $VOC_R$, as shown in Eq. (4).
$$Missing\ VOC_R = a[\text{CO}] + b[O_X] + c[isoprene_{initial}] + C_{backgound} \qquad (4)$$
where $O_X$ is defined as $O_3+NO_2$. [CO], $[O_X]$ and $[isoprene_{initial}]$ are concentrations
of tracers for anthropogenic emissions, secondary production and biogenic emissions,
respectively. $[isoprene_{initial}]$ represents the initial concentration of isoprene from
biogenic emissions that has not undergone any photochemical reactions, which is
calculated from observed isoprene and its photochemical products MVK and MACR





(Xie et al., 2008). $C_{background}$ indicates the background level of missing VOC$_R$. a, b,
c and $C_{backgound}$ are fitted coefficients by the multiple linear regression.
**2.5 Observation-based box model**

A zero-dimensional box model coupled with the Master Chemical Mechanism

(MCM) v3.3.1 chemical mechanism(Jenkin et al., 2003) was used to simulate the
photochemical production of RO$_X$ (RO$_X$=OH+HO$_2$+RO$_2$) radicals and O$_3$ during the
field campaign. The model was constrained by the observations of meteorological
parameters, photolysis frequencies, VOCs, NO, NO$_2$, O$_3$, CO, SO$_2$ and HONO. The
model runs were performed in a time-dependent mode with a time resolution of 1 hour
and a spin-up of four days. A 24-h lifetime was introduced for all simulated species,
including secondary species and radicals, to approximately simulate dry deposition and
other losses of these species (Lu et al., 2013;Wang et al., 2020b). Sensitivity tests show
that this assumed physical loss lifetime has a relatively small influence on RO$_X$ radicals
and ozone production rates.

Measured OVOCs such as HCHO, acetaldehyde and acetone were constrained in

the model and unmeasured OVOCs were simulated according to the photochemical
oxidation of NMHCs by OH radicals. RO$_2$, HO$_2$ and OH radicals were simulated by the
box model to calculate the net O$_3$ production rate (P(O$_3$)) and O$_3$ loss rate (L(O$_3$)) as
shown in Equations (5) and (6) as derived by Mihelcic et al. (2003)
$P(O_3) = k_{HO_2+NO}[HO_2][NO] + \sum_i(k_{RO_2+NO}^i[RO_2^i][NO]) - k_{OH+NO_2}[OH][NO_2] - L(O_3)$
$\hspace{20em}$ (5)
$L(O_3) = (\theta j(O^1D) + k_{OH+O_3}[OH] + k_{HO_2+O_3}[HO_2] + \sum_j(k_{alkene+O_3}^j[alkene^j])[O_3]$
$\hspace{20em}$ (6)
where θ is the fraction of O$^1$D from ozone photolysis that reacts with water vapor, and
i and j represent the number of species of RO$_2$ and alkenes, respectively.





## 3 Results and discussion

### 3.1 Quantification of missing VOC$_R$ during the campaign

**Figure 1** shows the time series of measured R$_{OH}$, calculated R$_{OH}$ according to all measured reactive gases, and missing VOC$_R$ (the gap between measured and calculated R$_{OH}$) in Guangzhou. By using PTR-ToF-MS, we measured many VOC species that were difficult before. Besides the NMHCs species with carbons less than 12, PTR-ToF-MS can also measure higher NMHCs with more carbons (C12–C20). With regard to OVOCs, not only common OVOC species including formaldehyde and C2-C4 carbonyls but also some N-containing OVOC species such as nitrophenol, methyl nitrophenol and several organic nitrates were measured. Thanks to these additional measured VOCs, the measured R$_{OH}$ was close to the calculated R$_{OH}$ within 20% in most periods. Nevertheless, there were still some days exhibiting remarkable missing VOC$_R$. The days with missing VOC$_R$ of more than 25% of total R$_{OH}$, namely high missing-VOC$_R$ days, are indicated by yellow background in **Fig. 1a**. The largest missing VOC$_R$ occurred on October 15$^{th}$, 16$^{th}$, 25$^{th}$ and 26$^{th}$, with average values of 16 s$^{-1}$. During the period of October 24$^{th}$ to 26$^{th}$, the total R$_{OH}$ was highest and the missing VOC$_R$ was also relatively high among all days. **Figure 1b** shows the contribution of different species classifications to total R$_{OH}$ during high missing–VOC$_R$ days. Inorganic species, NMHCs and OVOCs account for 34%, 13% and 14% of total R$_{OH}$, respectively, with missing VOC$_R$ accounting for 39%. The fraction of missing VOC$_R$ (39%) during the high missing–VOC$_R$ days is comparable to measurements in Los Angeles 2010 (Griffith et al., 2016) and in Seoul 2016 (Sanchez et al., 2021).

We evaluated the uncertainty of the missing VOC$_R$. The uncertainty of the R$_{OH}$ measurement was 15%. In addition, according to reports of Jet Propulsion Laboratory (Burkholder et al., 2020), reaction rate constants used for the calculation of R$_{OH}$ in Eq (3) have uncertainties of 5%–30%, depending on different species. We took these uncertainties into account when calculating R$_{OH}$, according to error propagation. As the




result, the uncertainties in the missing $VOC_R$ are 3.8 $s^{-1}$ and 5.2 $s^{-1}$ for the whole
measurement period and the high missing-$VOC_R$ days, respectively. The average
missing $VOC_R$ during the high missing-$VOC_R$ days is 12.3 $s^{-1}$, which is significantly
higher than the uncertainty of 5.2 $s^{-1}$, suggesting that the missing $VOC_R$ really exists
during the high missing-$VOC_R$ days.

**3.2 The sources of missing $VOC_R$**

To explore the sources of missing $VOC_R$ during the whole measurement period,

we investigated the correlation between missing $VOC_R$ and tracers characterizing
primary emissions (CO, $NO_X$ and NMHCs) and secondary production ($O_X \equiv O_3 + NO_2$
and formic acid). The correlation of missing $VOC_R$ with CO, reactivity of NMHCs
($NMHC_R$) and $NO_X$ is moderate, with correlation coefficient (R) in the range of 0.47–
0.56 (**Fig. 2a and b, and Fig. S1**) while there is no significant correlation of missing
$VOC_R$ with $O_X$ and formic acid (**Fig. 2c and Fig. S1**). Furthermore, there is no
significant correlation between missing $VOC_R$ and acetonitrile which is a tracer of
biomass burning (de Gouw et al., 2003;Wang et al., 2007) (**Fig. S1**), indicating that
biomass burning was not a major contributor to missing $VOC_R$ during this campaign.
In terms of the diurnal variation, the missing $VOC_R$ was higher in the morning (7:00–
10:00) and evening (18:00–22:00.) when the anthropogenic emissions, especially
vehicle exhaust were intensive, and was lower in the afternoon when the
photochemistry was most active (**Fig. 2d**). The diurnal profile of missing $VOC_R$ was
similar to those of CO, $NO_X$ and $NMHC_R$. In contrast, the diurnal profiles of secondary
species including $O_X$, formic acid and acetic acid, which peaked in the afternoon,
evidently differ from the diurnal profile of missing $VOC_R$ (**Fig. S2**). Further, we
investigated the influence of airmass aging on missing $VOC_R$. The ratio of ethylbenzene
to m,p-xylene was used to characterize the aging degree of air masses (De Gouw et al.,
2005;Yuan et al., 2013). A higher ratio of ethylbenzene to m,p-xylene corresponds to a
higher aging degree of air masses as the m,p-xylene has a larger reaction rate constant





than ethylbenzene when reacting with the major oxidant - OH radicals. As shown in **Fig. 2e**, missing $VOC_R$ decreases with the ratio of ethylbenzene to m,p-xylene. Given that secondary production generally increased with airmass aging, this result further demonstrates that missing $VOC_R$ was not caused by enhanced secondary production.

Given the larger missing $VOC_R$ level during the high missing- $VOC_R$ days, we focus on the high missing- $VOC_R$ days in the following analysis. During the high missing- $VOC_R$ days, the correlation coefficient for missing $VOC_R$ versus CO is 0.76 **(Fig. 3a)**, which is higher than that in the whole measurement period (0.56) shown in **Fig. 2a**. In addition, the correlation between missing $VOC_R$ and $O_X$ is weak with R=-0.25 during the high missing- $VOC_R$ days **(Fig. 3b)**. We then quantify the sources of missing $VOC_R$ during the high missing- $VOC_R$ days by applying MLR. The coefficient of determination ($R^2$) for the MLR is 0.68. As shown in **Fig. 3c,** anthropogenic emissions were the largest contributor to missing $VOC_R$, accounting for 70% of missing $VOC_R$. Secondary production, biogenic emissions and background contribution played a minor role in missing $VOC_R$ (13%, 7%, 10%, respectively). The parametric relationship between missing $VOC_R$ and relevant tracers established by MLR provides a valid approach to estimate the missing $VOC_R$ according to readily available gases including CO, $O_X$ and isoprene.

Although anthropogenic emissions are identified to be the major source of missing $VOC_R$, which species dominantly contribute to the missing $VOC_R$ remains unclear. A potential source is the unmeasured branched alkenes for their high reactivity, previously observed from vehicle exhaust (Nakashima et al., 2010) and gasoline evaporation emissions (Wu et al., 2015). Another possible source is emitted OVOCs with a more complex functional group that cannot be accurately measured. In addition, directly emitted semi-volatile and intermediate volatility organic compounds are also possible sources of missing $VOC_R$ (Stewart et al., 2021).

**3.3 The impact of missing $VOC_R$ on $O_3$ sensitivity regimes**

The reaction of OH with VOCs is key to the propagation and amplification of OH





radicals, thus determining the ozone production rate (Tonnesen and Dennis, 2000). The
box model was used to evaluate the impact of missing $VOC_R$ on the $O_3$ production rate
during high missing–$VOC_R$ days. In the base scenario, the box model was constrained
by all measured inorganic and organic gases but the missing $VOC_R$ was not considered.
To consider the missing $VOC_R$ in the box model, we increased all measured NMHC
species by a factor that can compensates for the missing $VOC_R$. In addition, we also try
adding a single VOC species to represent the missing $VOC_R$. Three typical VOC species
were added respectively, including n-pentane, ethylene and toluene. **Figure 4** shows the
simulated $P(O_3)$ for the base scenario and the one considering missing $VOC_R$. The
daytime average $P(O_3)$ under the scenario considering missing $VOC_R$ is a factor of 1.5-
4.5 for the results under the base scenario. The difference in added species has a large
effect on $P(O_3)$. Adding toluene causes a larger increase in $P(O_3)$ than adding n-pentane
or ethene, as toluene has a stronger ability to amplify the production of radicals. The
uncertainty in missing $VOC_R$ leads to 13-17% uncertainties in the threshold of $NO_X$ for
scenarios considering missing $VOC_R$.

$O_3$ precursor sensitivity depends on the dominant loss pathways of $RO_X$ radicals

($RO_X$=OH+HO_2+RO_2). $O_3$ production is $NO_X$-limited if the self-reaction of peroxy
radicals ($HO_2$ and $RO_2$) dominates the $RO_X$ sink, and VOC-limited if the reaction of
$NO_2$ with OH dominates (Kleinman et al., 1997;Kleinman et al., 2001). Accordingly,
the ratio of $RO_X$ sink induced by OH+$NO_2$ reaction to the total rate of the two $RO_X$
sinks, i.e., $L_N/Q$, is used to identify $O_3$ sensitivity regimes. $O_3$ production is $NO_X$-
limited if $L_N/Q$ is lower than 0.5, otherwise, it is VOC-limited (Kleinman et al., 1997).
$L_N/Q = \frac{k_{OH+NO_2}[OH][NO_2]}{k_{HO_2+RO_2}[HO_2][RO_2]+k_{HO_2+HO_2}[HO_2][HO_2]+k_{OH+HO_2}[OH][HO_2]+k_{OH+NO_2}[OH][NO_2]}$

(7)

As shown in **Fig. 5a**, under the base scenario, $L_N/Q$ remained at a stable and high

level (>0.9) during the daytime when photochemical production of ozone occurs,
indicating $O_3$ production was VOC-limited. Under the scenario considering missing
$VOC_R$, $L_N/Q$ decreased significantly regardless of which VOC species was added,
compared to the base scenario. Adding toluene caused the largest decrease in $L_N/Q$,





followed by adding all measured NMHC species, adding the alkane and adding the
alkene. It is worth noting that adding toluene and all measured NMHC species caused
the $L_N/Q$ to be close to 0.5 in the afternoon, indicating that the $O_3$ production shifted to
transitional or $NO_X$-limited regimes in these scenarios. **Fig. 5b** shows the changes in
radical sinks before and after considering missing $VOC_R$. All radical sinks including
self-reactions of peroxy radicals and $OH+NO_2$ reaction increased after considering
missing $VOC_R$. Nevertheless, the increased proportion of the self-reactions of peroxy
radicals was larger than that of $OH+NO_2$ reaction, leading to a decrease in $L_N/Q$ and
thus a shift toward $NO_X$-limited regime.

**Figure 5c** shows the dependence of daily peak $O_3$ concentrations on $NO_X$

concentrations, which was calculated by the box model for the base scenario and the
scenario considering missing $VOC_R$. The $NO_X$ concentration level corresponding to the
maximum of $O_3$ concentrations was determined. This $NO_X$ concentration level reflects
the threshold to distinguish between VOC-limited and $NO_X$-limited regimes. The larger
threshold of $NO_X$ represents a higher possibility of ozone production in $NO_X$ limited
regime. The threshold of $NO_X$ for the scenario considering missing $VOC_R$ is 46% higher
than for the base scenario. Note that the uncertainty in missing $VOC_R$ leads to 17%
uncertainty in the threshold of $NO_X$ for the scenario considering missing $VOC_R$. Overall,
**Fig. 5** suggests that omitting the missing $VOC_R$ will overestimate the degree of the
VOC-limited regime and thus overestimate the effect of VOCs abatement in reducing
ozone pollution, which in turn may mislead ozone control strategy.
**4    Conclusions**

Although many previous studies have reported that photochemical production

processes and biogenic emissions are important sources of missing $VOC_R$ (Lou et al.,
2010;Dolgorouky et al., 2012;Yang et al., 2017;Sanchez et al., 2021;Di Carlo et al.,
2004), we find that anthropogenic emissions may dominate the missing $VOC_R$ in urban
regions. In zero-dimensional box models and three-dimensional chemistry-transport
models, the input of VOCs emission information mainly contains well-studied simple-



structure alkanes, alkenes and aromatics, while those unmeasured/unknown VOC species have been neglected. This will lead to biases in quantifying ozone production and diagnosing ozone sensitivity regimes. Our study demonstrates that the ambient measurement of $R_{OH}$ at urban sites can provide quantification of missing $VOC_R$, which can be used in models to account for the missing $VOC_R$ from anthropogenic emissions. In addition, the parametric equation of missing $VOC_R$ versus CO developed here can be used to estimate missing $VOC_R$ according to CO measurements. Besides CO, other specific classes of hydrocarbons are also expected to be used as tracers for the development of the parametric equation. Further study should try to parse the specific sources of the missing $VOC_R$, e.g., whether the missing $VOC_R$ is from intermediate-volatility and semi-volatile organic compounds emitted from vehicles or whether it is from some other sources. Furthermore, future studies can focus on direct measurements of missing $VOC_R$ for various emission sources to develop a comprehensive emission inventory of missing $VOC_R$, which will help to improve $O_3$ pollution mitigation strategies.

## Acknowledgement

This work was supported by the National Natural Science Foundation of China (grant No. 42121004, 42275103, 42230701, 42175135). This work was also supported by Special Fund Project for Science and Technology Innovation Strategy of Guangdong Province (Grant No.2019B121205004).

## Competing interests

Two of the authors (Dr. Hang Su and Dr. Yafang Cheng) are members of the editorial board of ACP.

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






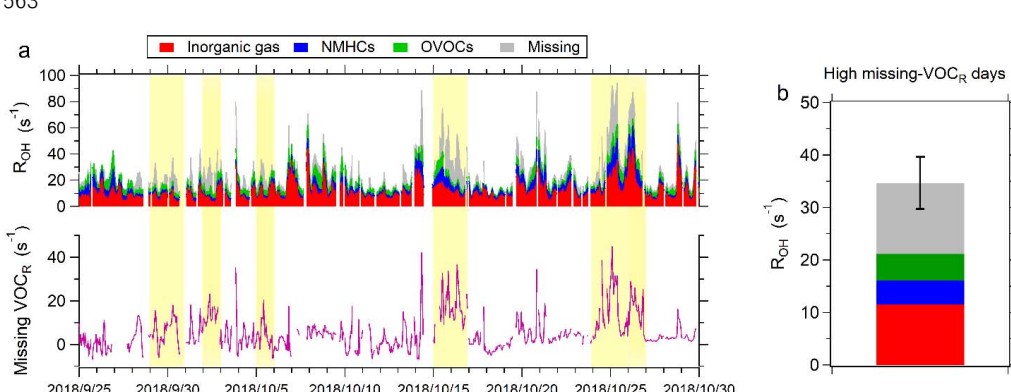

**Figure 1. The level of missing VOC_R during the measurements in Guangzhou.** (a)
Time series of measured $R_{OH}$ and calculated $R_{OH}$ from all measured reactive gases in
Guangzhou. Yellow background represents the high missing-VOC_R days with missing
VOC_R accounting for more than 30% of total $R_{OH}$. (b) Contributions of different
compositions to $R_{OH}$ in high missing-VOC_R days. The error bar represents standard
deviation of missing VOC_R.







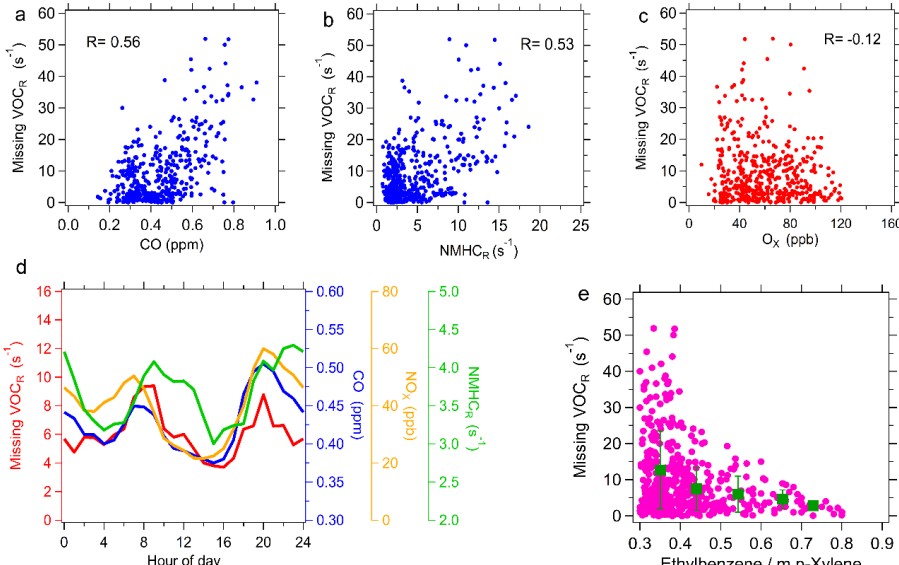

**Figure 2. Correlation of missing VOC$_R$ with major tracers during the whole measurement period.** (a-c) Correlation of missing VOC$_R$ with CO, OH reactivity of NMHCs (NMHC$_R$) and O$_X$. Each point represents hourly data. (d) Diurnal variations in missing VOC$_R$, CO, NO$_X$ and NMHCs. (e) The dependence of missing VOC$_R$ on ethylbenzene to m, p-xylene ratio.



580

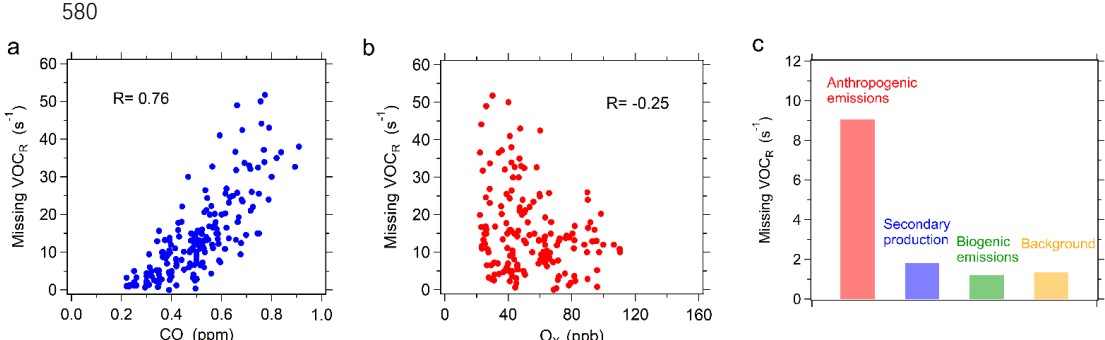

**Figure 3. The source apportionment of missing VOC$_R$ in high missing-VOC$_R$ days.**

(a) Correlation of missing VOC$_R$ with CO. (b) Correlation of missing VOC$_R$ with O$_X$.

In (a) and (b), each point represents hourly data. (c) Contributions of different sources

to missing VOC$_R$ according to the MLR.

585





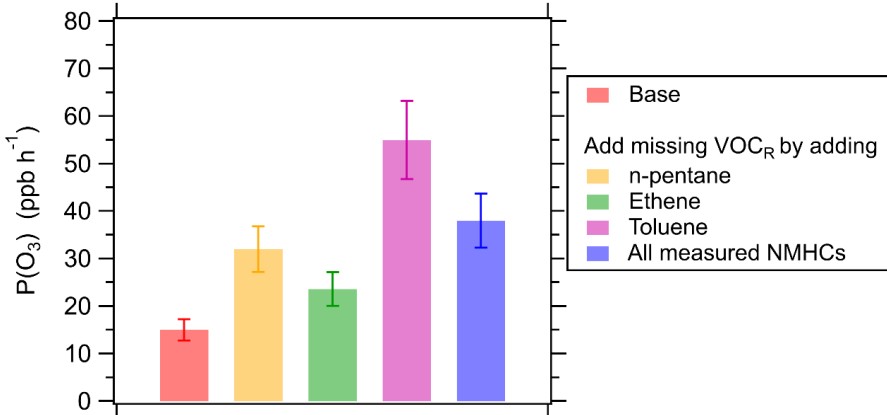

**Figure 4. Simulated daytime mean $P(O_3)$ for the base scenario (without missing VOC_R) and the scenario considering missing VOC_R, respectively, in high-missing VOC_R days.** The missing VOC_R is considered by adding individual species (n-pentane, ethene or toluene) or increasing all measured NMHCs to compensate for the missing VOC_R. The error bar represents standard deviation of $P(O_3)$ induced by the uncertainty of missing VOC_R.

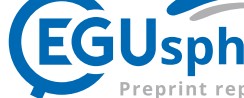

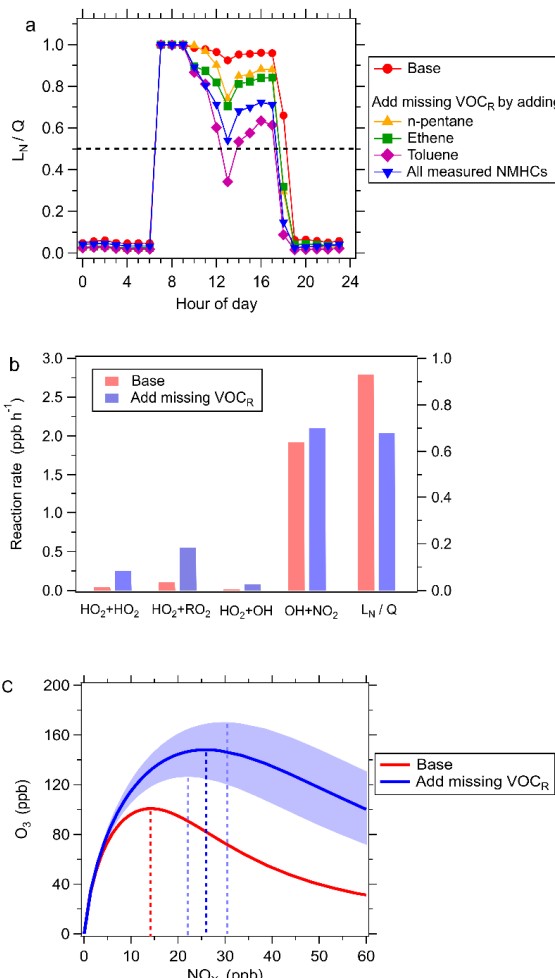

**Figure 5. The impact of missing VOC$_R$ on O$_3$ sensitivity for the high-missing VOC$_R$ days.** (a) Diurnal variations in L$_N$/Q for base scenario and the scenario considering missing VOC$_R$ (blue bar). The missing VOC$_R$ is considered by adding individual species (n-pentane, ethene or toluene) or increasing all measured NMHCs to fill the missing VOC$_R$. (b) The averages of radical sinks in the afternoon (12:00-18:00) for base scenario (red bar) and the scenario considering missing VOC$_R$ (blue bar) by increasing all measured NMHCs to fill the missing VOC$_R$. (c) Model simulated dependence of daily peak O$_3$ concentrations on daily mean NO$_X$ concentrations for base scenario (red curve) and the scenario considering missing VOC$_R$ (blue curve) by increasing all





measured NMHCs to fill the missing $VOC_R$. The dashed lines parallel to Y-axis represent the threshold of $NO_X$ levels to distinguish between VOC-limited and $NO_X$-limited regimes. The shaded area represents standard deviation induced by the uncertainty in missing $VOC_R$.