# Peer review of "A large role of missing volatile organic compounds reactivity"

_EGUsphere, 2023_

## Author Comment (AC1)

**Response to Referee #1's comments**

**General Comments:**

Interesting work with clear and significant conclusions, but requires more detail and explanation in some areas. For example, more context and clarity are required in the introduction and methods sections. and further experimental details to aid reproducibility. Some edits required to make the language used more fluent and precise.

**Specific Comments:**

Introduction:

Line 45 – "No one instrument can capture all VOCs out there and even when they can be measured there is information missing on identification and properties (Yuan et al., 2017; Wang et al., 2014)". This sentence needs expanding into a paragraph explaining what these species, instruments, and measurement methods are. Which VOC species can't currently be measured/haven't been measured and using which techniques?

Reply: Thanks for your suggestion. We have included this discussion accordingly.

Lines 46-61: Gas chromatograph–mass spectrometer/flame ionization detector (GC–MS/FID) can measure C2-C12 non-methane hydrocarbons (NMHCs) and C2-C6 oxygenated VOCs (OVOCs) while cannot measure NMHCs and OVOCs with larger carbon number (Wang et al., 2014). Proton-transfer-reaction time-of-flight mass spectrometer (PTR-ToF-MS) is able to measure a huge number of OVOCs and aromatics and several alkanes, but cannot measure most alkanes and alkenes, and cannot distinguish isomers (Yuan et al., 2017). The 2,4-dinitrophenylhydrazine (DNPH)/high performance liquid chromatography (HPLC) method can measure several carbonyls but cannot measure non-polar organic species (Wang et al., 2009). The two-dimensional GC is able to measure some intermediate-volatile and semi-volatile non-polar organics (Song et al., 2022). A lack of standard gases prevents these technologies from accurate quantification even if these technologies can identify more

VOC species. In general, many branched alkenes, OVOCs with complex functional groups, intermediate-volatile and semi-volatile organics and complex biogenic VOCs cannot currently be well quantified even if they can be identified by instruments.

Line 47 – "By now, emission inventories of VOCs used in air quality models only include the VOC species that can be measured". This sentence needs some clarity. Would suggest changing `by now` to `currently` and giving examples of which emissions inventories and VOC species are being discussed.

Reply: Thanks for your suggestion. We have modified it in the manuscript accordingly.

Lines 62-66: Currently, emission inventories used in air quality models such as the Community Emissions Data System (CEDS) emission inventory and the multi-resolution Emission Inventory for China (MEIC) only include the VOC species that can be measured such as some C1-C9 hydrocarbons and simple-structure OVOCs with small carbon number (<C6).

Line 70 – "The inclusion of the missing VOCR can help to improve the model performance in simulating photochemistry processes". Clarify which model.

Reply: Thanks for your suggestion. We have modified it in the manuscript accordingly. The model refers to box model and air quality models.

Lines 87-89: The inclusion of the missing $VOC_R$ can help to improve the performance of box model and air quality models in simulating photochemistry processes.

Line 90 – "Given that the missing VOCR accounts for a large part of total VOCR". This should probably read `could potentially account for`, or something similar, as this has not been determined yet.

Reply: Thanks for your suggestion. We have modified it accordingly.

Lines 109-113: Given that the missing $VOC_R$ could potentially account for a large part of total $VOC_R$, clearly clarifying the role of missing $VOC_R$ in determining ozone

precursor sensitivity is an urgent need for the diagnosis of ozone sensitivity regimes and formulation of an effective emission reduction roadmap.

Method:

More details required on the experimental procedure. For example, how and where the instruments were deployed. Were continuous measurements taken from 26th Sept to 30th Oct? Were the GC, PTR, and custom-built instrument run simultaneously?

Reply: Thanks for your suggestion. We have included this discussion accordingly.

Lines 121-128:

2.1 Overview of the measurement

The field campaign was conducted from 25 September to 30 October 2018 continuously at an urban site in downtown Guangzhou (113.2ºE, 23ºN). The sampling site was located on the ninth floor of a building on the campus of Guangzhou Institute of Geochemistry, Chinese Academy of Sciences, 25 m above the ground level. This site is primarily influenced by industrial and vehicular emissions. $R_{OH}$, VOCs, $NO_X$, $O_3$, HONO, $SO_2$, CO, photolysis frequencies, and meteorological factors were simultaneously measured during the measurement period.

PTR - Was this run in selected ion monitoring mode? If so, which reagent and product ions were selected? (Would suggest including a table of these in the SI.) Other missing details include drift tube pressure, temperature, and voltage, etc.

Reply: Thanks for your suggestion. We have modified it accordingly.

Lines 173-181: During the campaign, the PTR-ToF-MS automatically switched between $H_3O^+$ and $NO^+$ chemistry every 10–20 min. The $H_3O^+$ mode was used to measure OVOCs and aromatics while the $NO^+$ model was used to measure alkanes with more carbons (C8-C20). When running in the $H_3O^+$ ionization mode, the drift tube was at a temperature of 50 °C, a pressure of 3.8 mbar, and a voltage of 920 V, leading to an

operating E/N (E is the electric field, and N is the number density of the gas in the drift tube) ratio of 120 Td. When running in the $NO^+$ ionization mode, the drift tube was at a temperature of 50 ℃, a pressure of 3.8 mbar, and a voltage of 470 V, leading to an operating E/N ratio of 60 Td.

Include details of GC–MS/FID parameters.

Reply: Thanks for your suggestion. We have modified it accordingly.

Lines 161-170: After removal of water vapor, VOCs were trapped at -155 ℃ in a deactivated quartz capillary column (15 cm×0.53 mm ID) and a Porous Layer Open Tubular (PLOT) capillary column (15 cm×0.53 mm ID) for the MS channel and the FID channel, respectively. The system was calibrated weekly by TO-15 (Air Environmental Inc., USA) and PAMS gas standards (Spectra Gases Inc., USA). Detection limits for various compounds were in the range of 0.002–0.070 ppbv. A total of 56 NMHCs species were measured (**Table S1**). The time resolution of the measurement was 1 h. The uncertainties of VOC measurements by GC–MS/FID are in the range of 15 %–20 %. More details of this method can be found in previous studies (Wang et al., 2014;Yuan et al., 2012).

Line 119 – Can the correction curve be supplied in the SI?

Reply: Thanks for your suggestion. We have supplied the correction curve in the SI.

[Figure]

Figure S1. NO-correction experiments and fitting curves in Guangzhou in 2018 at different $R_{OH}$ of propene standard gas and mixture standard gas. The mixture standard gas used is the mixture PAMS (photochemical assessment monitoring stations) of 56 non-methane hydrocarbons (NMHCs; SpecialGas Ltd, USA).

Line 134 - `A total of 56 NMHCs species were measured`, provide information in SI.

Reply: Thanks for your suggestion. We have provided the information in SI.

**Table S1. VOC species measured in this study**

| Classes | VOC species |
|---|---|
| Alkane | ethane, propane, isobutane, n-butane, cyclopentane, isopentane, n-pentane, 2,2-dimethylbutane, 2,3-dimethylbutane, 2-methylpentane, 3-methylpentane, n-hexane, 2,4-dimethylpentane, methylcyclopentane, 2-methylhexane, cyclohexane, 2,3- |

| | dimethylpentane, 3-Methylhexane, 2,2,4-trimethylpentane, n-heptane, methylcyclohexane, 2,3,4-trimethylpentane, 2-methylheptane, 3-methyl Heptane, octane, n-nonane, n-decane, n-undecane, n-dodecane |
|---|---|
| Alkene | ethylene, propylene, trans-2-butene, 1-butene, cis-2-butene, 1,3-butadiene, 1-pentene, trans-2-pentene, isoprene, cis- 2-pentene, 1-hexene |
| Aromatic | benzene, ethylbenzene, m/p-xylene, o-xylene, styrene, n-propylbenzene, 3-ethyltoluene, 4-ethyltoluene, 1,3,5-trimethyl Benzene, 2-ethyltoluene, 1,2,4-trimethylbenzene, 1,2,3-trimethylbenzene, 1,3-diethylbenzene, 1,4-diethylbenzene, toluene |

Line 142 - `A total of 31 VOCs were calibrated using either gas or liquid standards`, provide information in SI.

Reply: Thanks for your suggestion. We have supplied this information in the SI.

Table S2. The 31 VOCs which were calibrated using either gas or liquid standards. The ion formula of these VOCs detected by PTR-ToF-MS and corresponding sensitivity are provided.

| VOC species | Ion formula | Sensitivity, cps/ppb |
|---|---|---|
| Formaldehyde | $CH_2OH^+$ | 1042 |
| Methanol | $CH_4OH^+$ | 629.3 |
| Acetonitrile | $C_2H_3NH^+$ | 3374 |
| Acetaldehyde | $C_2H_4OH^+$ | 2767 |

| | | |
|---|---|---|
| Ethanol | $C_2H_6OH^+$ | 99.23 |
| Acrolein | $C_3H_4OH^+$ | 4107 |
| Acetone | $C_3H_6OH^+$ | 4299 |
| Furan | $C_4H_4OH^+$ | 2544 |
| Isoprene | $C_5H_8H^+$ | 1888 |
| MVK+MACR | $C_4H_6OH^+$ | 3868 |
| MEK | $C_4H_8OH^+$ | 4467 |
| Benzene | $C_6H_6H^+$ | 3151 |
| 2-Pentanone | $C_5H_{10}OH^+$ | 4510 |
| Toluene | $C_7H_8H^+$ | 3978 |
| Phenol | $C_6H_6OH^+$ | 4076 |
| Furfural | $C_5H_4O_2H^+$ | 7460 |
| Methyl Isobutyl Ketone | $C_6H_{12}OH^+$ | 3988 |
| Styrene | $C_8H_8H^+$ | 4289 |
| xylene | $C_8H_{10}H^+$ | 4241 |
| Cresol | $C_7H_8OH^+$ | 4299 |
| Trimethylbenzene | $C_9H_{12}H^+$ | 4413 |
| Naphthalene | $C_{10}H_8H^+$ | 5117 |
| a-Pinene | $C_{10}H_{16}H^+$ | 2332 |
| Formic acid | $CH_2O_2H^+$ | 856.6 |
| Acetic acid | $C_2H_4O_2H^+$ | 1711 |
| Propionic acid | $C_3H_6O_2H^+$ | 2072 |
| Butyric acid | $C_4H_8O_2H^+$ | 2358 |
| Pyrrole | $C_4H_5NH^+$ | 2842 |
| Formamide | $CH_3NOH^+$ | 2871 |
| Acetamide | $C_2H_5NOH^+$ | 3992 |

Line 143 - `For other measured VOCs`, provide details in SI.

Reply: Thanks for your suggestion. The other measured VOCs include 128 VOC species. The detailed information can be seen in Wu et al. (2020) and all VOC species measured by PTR-ToF-MS were provided in table S4 of that article.

Lines 190-193: By this method, PTR-ToF-MS can additionally measure 128 VOCs which were included in the analysis of this study. The detailed information for this method can be found in Wu et al. (2020) and all VOC species measured by PTR-ToF-MS were provided in table S4 of that article.

Results and Discussion:

Line 206 - How many extra species were measured by PTR?

Reply: Thanks for your suggestion. By using PTR-ToF-MS, we measured 159 VOCs and 128 of them were difficult to be measured before. We have provided this information accordingly.

Lines 265-271: By using GC-MS/FID, we measured 56 NMHCs. By using PTR-ToF-MS, we measured 159 VOCs and 128 of them were difficult to be measured before. Besides the alkanes with carbons less than 12, PTR-ToF-MS can also measure alkanes with more carbons (C12–C20). With regard to OVOCs, not only common OVOC species including formaldehyde and C2-C4 carbonyls but also carbonyls with more carbons (C5-C10) and some N-containing OVOC species such as nitrophenol and methyl nitrophenol were measured by PTR-ToF-MS.

**Technical Corrections:**

Line 53 – "The measurement of total OH reactivity (ROH) provides an effective approach to quantify the total amount of reactive gases in terms of reacting with OH radicals." I think this sentence could be reworded to be clearer and more concise.

Reply: Thanks for your suggestion. We have modified it accordingly.

Lines 70-71: The total OH reactivity ($R_{OH}$), which can be directly measured, is an index for evaluating the amounts of reductive pollutants in terms of ambient OH loss.

Line 166 – "The multiple linear regression (MLR) have been successfully applied to quantify the sources of air pollutants (Li et al., 2019; Yang et al., 2016a)." Grammar – `have` should be `has`.

Reply: Thanks for your suggestion. We have modified it accordingly.

Lines 211-212: The Multiple Linear Regression (MLR) has been successfully applied to quantify the sources of air pollutants (Li et al., 2019; Yang et al., 2016a).

Line 176 – "Calculated from observed isoprene and its photochemical products MVK and MACR". Define acronyms.

Reply: Thanks for your suggestion. We have modified it accordingly.

Lines 222-223: calculated from observed isoprene and its photochemical products methyl vinyl ketone (MVK) and methacrolein (MACR).

Line 261 - ` Given the larger missing VOCR level during the high missing- VOCR days, we focus on the high missing- VOCR days in the following analysis.` This line seems unnecessary and repetitive.

Reply: Thanks for your suggestion. We have removed this sentence.

Line 290 – Compensate rather than compensates.

Reply: Thanks for your suggestion. We have modified it.

Define units for all equations and variables where appropriate throughout.

Reply: Thanks for your suggestion. We have defined units for all equations and variables in the SI.

Lines 128-129: The units of all parameters used in this study is shown in table S3.

Table S3. The units of variables used in this study.

| Variables | Units |
|-----------|-------|
| $R_{OH}$ | $s^{-1}$ |
| $k_{OH+Xi}$ | $ppb^{-1}\,s^{-1}$ |
| $[X_i]$ | ppb |
| $VOC_R$ | $s^{-1}$ |
| $Missing\ VOC_R$ | $s^{-1}$ |
| $C_{backgound}$ | $s^{-1}$ |
| $P(O_3)$ | $ppb\,h^{-1}$ |
| $L(O_3)$ | $ppb\,h^{-1}$ |
| $j(O^1D)$ | $s^{-1}$ |
| $L_N/Q$ | unitless |

Some of the details of how the box model was run from section 3.3 might be more appropriate in the methods section than in the results.

Reply: Thanks for your suggestion. We have moved the details of how the box model was run to the methods section 2.6.

Lines 249-260:

    The box model was used to evaluate the impact of missing $VOC_R$ on the $O_3$ production rate. In the base scenario, the box model was constrained by all measured inorganic and organic gases but the missing $VOC_R$ was not considered. To consider the missing $VOC_R$ in the box model, we additionally increased the concentration of NMHCs to exactly compensate for the missing $VOC_R$ by multiplying a factor, on the basis of measured NMHC concentrations. We simulated four scenarios by increasing the concentration of: (1) n-pentane, (2) ethylene, (3) toluene, (4) all measured 56 NMHCs. For the scenario of increasing all 56 NMHCs, concentrations of 56 NMHC

species were increased by multiplying the same factor. Given that the $VOC_R$ of unconstrained secondary products increases with the increase in the concentration of NMHCs, several attempts of different values are needed to determine the increasing factor.

**References:**

Song, K., Gong, Y., Guo, S., Lv, D., Wang, H., Wan, Z., Yu, Y., Tang, R., Li, T., Tan, R., Zhu, W., Shen, R., and Lu, S.: Investigation of partition coefficients and fingerprints of atmospheric gas- and particle-phase intermediate volatility and semi-volatile organic compounds using pixel-based approaches, Journal of Chromatography A, 1665, 462808, https://doi.org/10.1016/j.chroma.2022.462808, 2022.

Wang, H., Zhang, X., and Chen, Z.: Development of DNPH/HPLC method for the measurement of carbonyl compounds in the aqueous phase: applications to laboratory simulation and field measurement, Environmental Chemistry, 6, 389-397, https://doi.org/10.1071/EN09057, 2009.

Wang, M., Zeng, L., Lu, S., Shao, M., Liu, X., Yu, X., Chen, W., Yuan, B., Zhang, Q., and Hu, M.: Development and validation of a cryogen-free automatic gas chromatograph system (GC-MS/FID) for online measurements of volatile organic compounds, Anal. Methods, 6, 9424-9434, 2014.

Wu, C., Wang, C., Wang, S., Wang, W., Yuan, B., Qi, J., Wang, B., Wang, H., Wang, C., and Song, W.: Measurement report: Important contributions of oxygenated compounds to emissions and chemistry of volatile organic compounds in urban air, Atmos. Chem. Phys., 20, 14769-14785, 2020.

Yuan, B., Chen, W., Shao, M., Wang, M., Lu, S., Wang, B., Liu, Y., Chang, C.-C., and Wang, B.: Measurements of ambient hydrocarbons and carbonyls in the Pearl River Delta (PRD), China, Atmos. Res., 116, 93-104, 2012.

Yuan, B., Koss, A. R., Warneke, C., Coggon, M., Sekimoto, K., and de Gouw, J. A.: Proton-Transfer-Reaction Mass Spectrometry: Applications in Atmospheric Sciences, Chemical Reviews, 117, 13187-13229, 10.1021/acs.chemrev.7b00325, 2017.

---

## Author Comment (AC2)

**Response to Referee #2's comments**

This paper presents an interesting analysis for understanding missing VOC sources in urban areas. The science and the methodology are sound; however, the paper itself lacks depth. It seems as if the authors were hesitant to list out too much information within the manuscript. It also assumes any reader has the exact background knowledge to follow all arguments so the authors don't elaborate. While it is true readers can check the references themselves, and should, it would be nice if the authors could share tangible evidence from the sources that support their work. A paper should not only introduce a new idea or method but also be written in a way that subsequent papers can test the method for themselves and apply it to other data sets. As it is currently written that is not possible. This work definitely should be published but the authors need to go over the manuscript and hone the message. I suggest this paper is accepted with minor revisions because no additional analysis needs to be done but share a clearer message for what has already been done.

Reply: Many thanks for your comprehensive and valuable comments. According to your suggestion, we have extended the discussion in the manuscript. In the introduction, we provided more information about the current shortcoming of VOC measurements to highlight the importance of $VOC_R$ measurements. In the method, we provide more detailed introduction of GC-MS/FID, PTR-ToF-MS and the correction of NO interference on $R_{OH}$ measurement. In addition, we provided the detailed information about how to consider the missing $VOC_R$ in the box model. In the results of discussion, we provided the fitted coefficients of the MLR and improve relevant figures.

**Specific Comments**

Lines 90-91: You mention that missing $VOC_R$ is a large part of total $VOC_R$ but you don't give numbers or examples for the reader. You mention suburban sources in reference to $R_{OH}$ (Line 76) but not for $VOC_R$

Reply: Thanks for our suggestions. In different regions, the missing $VOC_R$ accounted for 10-75% of total $R_{OH}$. Given that total $VOC_R$ is one part of total $R_{OH}$, missing $VOC_R$ would account for a larger percentage of total $VOC_R$ (>10%-75%).

Lines 94-95: Given that total $VOC_R$ is one part of total $R_{OH}$, missing $VOC_R$ would account for a larger percentage of total $VOC_R$ (>10%-75%).

Line 127: Could you show how a $NO_x$ correction is applied? Perhaps in the supplement. What does "carefully applied" mean?

Reply: Thanks for your suggestion. We have depicted how the $NO_X$ correction curve is obtained and how it is applied to correct measured $R_{OH}$ in the supplement.

Lines 145-147: A correction curve was acquired from these NO interference experiments, which can be used to correct the $R_{OH}$ thanks to the simultaneous measurement of ambient NO concentrations (Supplementary information S1; Fig. S1). Lines 28-38 in supplementary information:

**S1 The correction of NO interference on $R_{OH}$ measurements**

The NO-correction experiments were conducted by introducing given amounts of VOC standard gases into the reactor. Different levels of NO were injected into the reactor and the difference between "measured" $R_{OH}$ and true $R_{OH}$ increased as the NO concentration increased. Here, the difference between "measured" $R_{OH}$ and true $R_{OH}$ is defined as $\delta R_{OH}$. Then, a correction curve was fitted between the $\delta R_{OH}$ and NO concentrations. Several standard gases (propene and PAMS mixture) and different levels of base reactivity (from 30 to 90 $s^{-1}$) have been tried and the curve was quite consistent for all tested gases, as shown in Fig. S. According to this correction curve and ambient NO concentrations, we calculated the $\delta R_{OH}$ which was used to correct the measured $R_{OH}$.

[Figure]

Figure S1. NO-correction experiments and fitting curves in Guangzhou in 2018 at different $R_{OH}$ of propene standard gas and mixture standard gas. The mixture standard gas used is the mixture PAMS (photochemical assessment monitoring stations) of 56 non-methane hydrocarbons (NMHCs; SpecialGas Ltd, USA).

Line 177 – 178: Why don't use you include values for a, b, c and $C_{background}$ in your paper? A main point in your conclusion is how this is new method to use the CO with $VOC_R$ to get at anthropogenic missing fraction but then you don't show any concrete numerical examples using this new method.

Reply: Thanks for your suggestion. We have provided the values for a, b, c and $C_{background}$ in the manuscript accordingly.

Lines 329-330: The fitted coefficient a is 0.031 $s^{-1}$ $ppb^{-1}$, b is 0.012 $s^{-1}$ $ppb^{-1}$, c is 1.8 $s^{-1}$ $ppb^{-1}$ and $C_{background}$ is 1.3 $s^{-1}$.

Lines 211-213: Did you ever have negative $VOC_R$? In other words, observations higher than the calculated missing $VOC_R$? It looks like you have some periods no visible gray in the figure for missing VOC, if negligible but still some 'missing' also good to point out when that happens as well as the general 20% since that is impressive given the issues you introduce in the beginning of the paper from previous work.

Reply: Thanks for your suggestions. Actually, the missing $VOC_R$ was negative during some periods. This is probably due to the uncertainty in the measurements of $R_{OH}$ and reactive gases. The negative missing $VOC_R$ primarily occurred in the afternoon (12:00–17:00) when the photochemistry was most active. Most of the negative missing $VOC_R$ values were larger than -5 $s^{-1}$.

Lines 273-276: In some periods the missing $VOC_R$ was negative, which is probably due to the uncertainty in the measurements of $R_{OH}$ and reactive gases. The negative missing $VOC_R$ primarily occurred in the afternoon (12:00–17:00) when the photochemistry was most active.

Line 256: Include the reaction rate constants used

Reply: Thanks for your suggestion. We have included the reaction rate constants.

Lines 319-322: A higher ratio of ethylbenzene to m,p-xylene corresponds to a higher aging degree of air masses as the m, p-xylene has a larger reaction rate constant ($18.9\times10^{-12}$ $cm^3$ $molecule^{-1}s^{-1}$) than ethylbenzene ($7.0\times10^{-12}$ $cm^3$ $molecule^{-1}s^{-1}$) when reacting with the major oxidant - OH radicals.

Line 293: By what factor did you increase all the NMHC? Were they all increased the same amount? What was the process here?

Reply: Under the base scenario, the measured $VOC_R$ of all 56 NMHCs are 4.6 $s^{-1}$. To consider the missing $VOC_R$ (13 $s^{-1}$) in the model, concentrations of the 56 NMHCs were increased by a factor of 1.9, leading to an additional increase in $VOC_R$ of both NMHCs and unconstrained secondary products, which exactly compensated for the missing $VOC_R$. Given that the $VOC_R$ of unconstrained secondary products increases with the increase in the concentration of NMHCs, several attempts of different values are needed to determine the increasing factor.

Lines 249-260: The box model was used to evaluate the impact of missing $VOC_R$ on the $O_3$ production rate. In the base scenario, the box model was constrained by all measured inorganic and organic gases but the missing $VOC_R$ was not considered. To consider the missing $VOC_R$ in the box model, we additionally increased the concentration of NMHCs to exactly compensate for the missing $VOC_R$ by multiplying

a factor, on the basis of measured NMHC concentrations. We simulated four scenarios by increasing the concentration of: (1) n-pentane, (2) ethylene, (3) toluene, (4) all measured 56 NMHCs. For the scenario of increasing all 56 NMHCs, concentrations of 56 NMHC species were increased by multiplying the same factor. Given that the $VOC_R$ of unconstrained secondary products increases with the increase in the concentration of NMHCs, several attempts of different values are needed to determine the increasing factor.

Lines in 350-357: The setting of model simulations for different scenarios are depicted in Section 2.6. Under the base scenario, on average the measured $VOC_R$ of n-pentane, ethylene, toluene and all 56 NMHCs are 0.14 $s^{-1}$, 0.53 $s^{-1}$, 0.60 $s^{-1}$ and 4.6 $s^{-1}$ respectively. To consider the missing $VOC_R$ (on average of 13 $s^{-1}$) in the model, four scenarios were simulated by additionally increasing n-pentane, ethylene, toluene and 56 NMHCs by a factor of 70, 16, 13.3 and 1.9, respectively. These increasing factors led to an additional increase in $VOC_R$ of both NMHCs and unconstrained secondary products, which exactly compensated for the missing $VOC_R$.

Section 3.3: The writing is unclear about the sensitivity studies. Were the individual VOC species (represented by the 3 examples) and the "all measured NMHC" done together or 4 different sensitivity studies? I'm assuming it was 4 different model runs but as written that isn't apparent and it sounds like they were done all together. It wasn't until looking at the figure it seemed like 4 runs. In particular, line 293 "and the one considering" suggests one model run which means you didn't look at the impact of each species. Were the individual species taken from the measured results? Where did those numbers come from?

Reply: Thanks for your suggestions. (1) We simulated four different scenarios, rather than they were done all together. (2) "and the one considering" has been changed into "and the scenarios considering". (3) To consider the missing $VOC_R$ in the box model, we additionally increased the concentration of NMHCs (individual species or all NMHCs) to exactly compensate for the missing $VOC_R$ by multiplying a factor, on the basis of measured NMHC concentrations.

We have modified the corresponding sentences in the manuscript to make the meaning more clearly.

Lines 249-260: The box model was used to evaluate the impact of missing $VOC_R$ on the $O_3$ production rate. In the base scenario, the box model was constrained by all measured inorganic and organic gases but the missing $VOC_R$ was not considered. To consider the missing $VOC_R$ in the box model, we additionally increased the concentration of NMHCs to exactly compensate for the missing $VOC_R$ by multiplying a factor, on the basis of measured NMHC concentrations. We simulated four scenarios by increasing the concentration of: (1) n-pentane, (2) ethylene, (3) toluene, (4) all measured 56 NMHCs. For the scenario of increasing all 56 NMHCs, concentrations of 56 NMHC species were increased by multiplying the same factor. Given that the $VOC_R$ of unconstrained secondary products increases with the increase in the concentration of NMHCs, several attempts of different values are needed to determine the increasing factor.

Lines in 350-357: The setting of model simulations for different scenarios are depicted in Section 2.6. Under the base scenario, on average the measured $VOC_R$ of n-pentane, ethylene, toluene and all 56 NMHCs are 0.14 $s^{-1}$, 0.53 $s^{-1}$, 0.60 $s^{-1}$ and 4.6 $s^{-1}$ respectively. To consider the missing $VOC_R$ (on average of 13 $s^{-1}$) in the model, four scenarios were simulated by additionally increasing n-pentane, ethylene, toluene and 56 NMHCs by a factor of 70, 16, 13.3 and 1.9, respectively. These increasing factors led to an additional increase in $VOC_R$ of both NMHCs and unconstrained secondary products, which exactly compensated for the missing $VOC_R$.

Line 358: **Figure 4** shows the simulated $P(O_3)$ for the base scenario and the scenarios considering missing $VOC_R$.

Lines 347-348: in regards to the parametric equation "developed here" do you mean being able to separate them all out? It isn't just "versus CO" according to equation 4 so this is misleading.

Reply: Thanks for your suggestion. We have modified this sentence to avoid misleading.

Lines 411-413: In addition, the parametric equation of missing $VOC_R$ derived from MLR method (Eq (4)) here can be used to estimate missing $VOC_R$ according to measurements of CO, $O_X$ and isoprene.

Lines 348-350: "are also expected" doesn't make sense in the context

Reply: Thanks for your suggestion. We have removed this sentence.

Figure Comments:

Figure 1: Why are a and b blue but c red? If for primary vs secondary that isn't referenced in the caption so is irrelevant since it doesn't match the color scheme in c or d. For example in d, missing VOC is red but in c it would be secondary sources. In e, what are the green squares? You don't reference them anywhere. Why are the circles magenta/pink? Not necessary and detracting.

Reply: Thanks for your suggestion. We have changed the color to make consistent. In e, we changed the green squares to red squares, which represents the mean values of missing $VOC_R$ in different ranges of ethylbenzene/m,p-xylene with classification width of 0.1.

[Figure]

**Figure 2. Correlation of missing $VOC_R$ with major tracers during the whole measurement period.** (a-c) Correlation of missing $VOC_R$ with CO, OH reactivity of NMHCs ($NMHC_R$) and $O_X$. Each point represents hourly data. (d) Diurnal variations

in missing $VOC_R$, CO, $NO_X$ and NMHCs. (e) The dependence of missing $VOC_R$ on ethylbenzene to m, p-xylene ratio. The red squares indicate the mean values of missing $VOC_R$ in different ranges of ethylbenzene/m,p-xylene with classification width of 0.1, and the error bars represent standard deviation.

Figure 2: Again the different colors for a and b seem unnecessary and then don't match c and are in fact opposite. CO = anthropogenic but it is blue and red in a and c, respectively.

Reply: Thanks for your suggestion. We think it is no need to provide the panel b (missing $VOC_R$ vs $O_X$) because the correlation between missing $VOC_R$ and $O_X$ is poor and this information has been provided in Fig. 2c. Thus, we have removed the panel b in the updated manuscript. In addition, we change the color of the first bar in panel c to be blue to make it consistent with panel a.

[Figure]

**Figure 3. The source apportionment of missing $VOC_R$ in high missing-$VOC_R$ days.** (a) Correlation of missing $VOC_R$ with CO. Each point represents hourly data. (b) Contributions of different sources to missing $VOC_R$ according to the MLR.

Figures 4 and 5: Nice use of colors here that tie the idea together.

Figure 5: there is no blue bar a but that is referenced in the caption. Also for a, it would be nice to note the dashed line represents the $NO_x$ vs VOC limited regimes.

Reply: Thanks for your suggestion. We have modified it accordingly.

The caption of Fig. 5: (a) Diurnal variations in $L_N/Q$ for the base scenario and the scenarios considering missing $VOC_R$. The missing $VOC_R$ is considered by adding individual species (n-pentane, ethene or toluene) or increasing all measured NMHCs to

fill the missing $VOC_R$. The dashed line represents the threshold value of $L_N/Q$ that distinguishes VOC-limited and $NO_X$-limited regimes.

**Technical Corrections**

Line 57: Indented but shouldn't be

Reply: Thanks. We have deleted the indent.

Line 68 and elsewhere: Perhaps a personal preference but the oxford comma can be very useful with complicated lists in sentences

Reply: Thanks. We have used the oxford comma for complicated lists in sentences.

Line 120: "thank to the" should be "thank**s** to the"

Reply: Thanks. We have revised it.

Line 131 and elsewhere: Be sure to have a space between a ) and the next word

Reply: Thanks. We have revised it.

Line 140: "cannot" is the more common spelling for this

Reply: Thanks. We have revised it.

Line 166: "The multiple linear regression (MLR) have" is awkward and incorrect tense. Perhaps something like "Multiple Linear Regression (MLR) has been"

Reply: Thanks. We have revised it.

Line 228: "As the" should be "As a"

Reply: Thanks. We have revised it.

Line 247: no period between 22:00 and )

Reply: Thanks. We have revised it.

Line 256: The wording of "higher aging degree of air masses" is awkward. Perhaps "higher degree of aging air masses" or "higher degree of air mass aging" based on what you write below at 259

Reply: Thanks. We have revised it.

Line 287: and elsewhere: Be consistent with – and spacing, sometimes an extra space and sometimes not

Reply: Thanks. We have revised it.

---

## Author Response (AR2)

We greatly appreciate the time and efforts that the Editor spent in guiding the review of our manuscript. The comments are really thoughtful and helpful to improve the quality of our paper. We have addressed each comment below, with the comment in black text, our response and relevant manuscript changes in blue text.

Technical corrections:

Heading, section 2.1: spelling of 'measurement'

Reply: Thanks. We have revised it.

Line 121: 2.1 Overview of the measurement

Line 366. It would be slightly clearer as. "The fitted coefficients are as follows: a is"

Reply: Thanks. We have revised it.

Line 328-329: The fitted coefficients are as follows: a is $0.031$ s$^{-1}$ ppb$^{-1}$, b is $0.012$ s$^{-1}$ ppb$^{-1}$, c is $1.8$ s$^{-1}$ ppb$^{-1}$ and $C_{background}$ is $1.3$ s$^{-1}$.

With the next file upload request, please include a section "Competing interests" (prior to the Acknowledgements) with the following text: At least one of the (co-)authors is a member of the editorial board of Atmospheric Chemistry and Physics.

Reply: Thanks. We have revised it.

Lines 418-420:

**Competing interests**

At least one of the (co-)authors is a member of the editorial board of Atmospheric Chemistry and Physics.